# Unveiling the Venom Composition of the Colombian Coral Snakes *Micrurus helleri*, *M. medemi*, and *M. sangilensis*

**DOI:** 10.3390/toxins15110622

**Published:** 2023-10-24

**Authors:** Ariadna Rodríguez-Vargas, Adrián Marcelo Franco-Vásquez, Janeth Alejandra Bolívar-Barbosa, Nohora Vega, Edgar Reyes-Montaño, Roberto Arreguín-Espinosa, Alejandro Carbajal-Saucedo, Teddy Angarita-Sierra, Francisco Ruiz-Gómez

**Affiliations:** 1Grupo de Investigación en Proteínas, Departamento de Química, Faculty of Sciences, Universidad Nacional de Colombia, Bogotá 11001, Colombianavegac@unal.edu.co (N.V.); eareyesm@unal.edu.co (E.R.-M.); 2Grupo de Investigación en Animales Ponzoñosos y sus Venenos, Dirección de Producción, Instituto Nacional de Salud, Bogotá 111321, Colombia; tgangaritas@unal.edu.co (T.A.-S.); fruiz@ins.gov.co (F.R.-G.); 3Departamento de Química de Biomacromoléculas, Instituto de Química, Universidad Nacional Autónoma de México, Coyoacán, Ciudad de México 04510, Mexicoarrespin@unam.mx (R.A.-E.); 4Laboratorio de Herpetología, Facultad de Ciencias Biológicas, Universidad Autónoma de Nuevo León, San Nicolás de los Garza 66450, Mexico; alejandro.carbajals@uanl.edu.mx; 5Grupo de investigación Biodiversidad para la Sociedad, Escuela de pregrados, Dirección Académica, Universidad Nacional de Colombia sede de La Paz, Cesar 22010, Colombia

**Keywords:** *Micrurus* sp., elapid venom, proteomic studies, cytotoxicity, biological activities

## Abstract

Little is known of the biochemical composition and functional features of the venoms of poorly known Colombian coral snakes. Here, we provide a preliminary characterization of the venom of two Colombian endemic coral snake species, *Micrurus medemi* and *M. sangilensis*, as well as Colombian populations of *M. helleri*. Electrophoresis and RP-HPLC techniques were used to identify venom components, and assays were conducted to detect enzyme activities, including phospholipase A_2_, hyaluronidase, and protease activities. The median lethal dose was determined using murine models. Cytotoxic activities in primary cultures from hippocampal neurons and cancer cell lines were evaluated. The venom profiles revealed similarities in electrophoretic separation among proteins under 20 kDa. The differences in chromatographic profiles were significant, mainly between the fractions containing medium-/large-sized and hydrophobic proteins; this was corroborated by a proteomic analysis which showed the expected composition of neurotoxins from the PLA_2_ (~38%) and 3FTx (~17%) families; however, a considerable quantity of metalloproteinases (~12%) was detected. PLA_2_ activity and protease activity were higher in *M. helleri* venom according to qualitative and quantitative assays. *M. medemi* venom had the highest lethality. All venoms decreased cell viability when tested on tumoral cell cultures, and *M. helleri* venom had the highest activity in neuronal primary culture. These preliminary studies shed light on the venoms of understudied coral snakes and broaden the range of sources that could be used for subsequent investigations of components with applications to specific diseases. Our findings also have implications for the clinical manifestations of snake envenoming and improvements in its medical management.

## 1. Introduction

Snakebite is an unresolved public health issue in Central and South America [1,2]. An average of ca. 57,500 snakebites occur per year in this region (6.34 cases per 100,000 inhabitants), resulting in almost 370 deaths (0.037/100,000 inhabitants) and a case fatality rate near 0.6% for the entire region [3]. Among South American countries, Colombia ranks third in the number of snakebites (ca. 4500 per year) and sixth in snakebite incidence (ca. 9.1/100,000 inhabitants), and 1.3% of the envenomations are caused by coral snakes of the genus *Micrurus* [3,4,5]. Approximately 56% of the patients envenomated by coral snakes exhibit serious life-threatening clinical manifestations, including respiratory muscle weakness; many of these patients need mechanical ventilation, which requires strict monitoring in an intensive care unit [5,6]. In addition, the onset of symptoms from *Micrurus* bites has a variable latency period ranging from 2 h in infants to 4–14 h for adolescents and adults [5].

Symptoms of envenomation induced by coral snakes include palpebral ptosis, ophthalmoplegia, inability to swallow, and paralysis of the larynx, neck, and skeletal muscles, including the diaphragm. These symptoms are the consequence of neuromuscular blockage at the motor synapsis [7,8], which is mainly induced by three-finger toxins (3FTx) and neurotoxic phospholipases A_2_ (PLA_2_) [9,10]. A total of 19 toxin families, including proteases, oxidases, and growth factors, have been documented in the venoms of *Micrurus* [11,12], and their specific roles in envenomation have not yet been clarified.

The complex geographic characteristics of Colombia, including the Andes, Chocó, and Amazon ecoregions, contribute to its high biodiversity [13,14,15]; however, this also poses a challenge for assessing the identity of the coral snake species responsible for envenomation. More than 30 species of coral snakes have been documented in Colombia, and Colombia’s coral snakes represent 35% of the total diversity of coral snake species in the Americas [16]. The cryptic behavior, semi-fossorial habits, and specialized microhabitats of *Micrurus* species [17] contribute to the low incidence of snakebites caused by this group. Systematic effort is needed to characterize the biochemical and functional composition of the Colombian coral snake venoms to better understand the pathophysiology of envenomation and aid the development of improved antivenom therapies for treating snakebites caused by *Micrurus* species.

Our goal is to present the first biochemical characterization of three poorly known coral snake species of the Colombian lowlands. *Micrurus helleri* (Linnaeus, 1758), formerly *M. lemniscatus helleri*, is a tricolored, medium-sized coral snake with a total length of 37.9–95.5 cm. It inhabits the eastern versant of the Cordillera Oriental, Orinoquía, and Amazon ecoregions in low mountain humid forest, disturbed areas, and even urban zones up to 1000 m above sea level (hereafter masl) [18]. It is considered a medically significant coral snake species. *Micrurus medemi* (Roze, 1967), commonly known as the Villavicencio coral snake, is a small- to medium-sized coral snake, ranging from 45 to 70 cm in total length, with a tricolored pattern of black, red, and yellow rings and heavily melanized red rings [18]. This species inhabits the eastern versant of the Cordillera Oriental and has only been documented in the Meta department. It is a common coral snake in urban and peri-urban areas of Villavicencio and adjacent municipalities of Acacias, Guayabetal, Manzanares, and Restrepo from 250 to 1577 masl [19,20,21]. *Micrurus sangilensis* (Nicéforo Maria, 1942) is restricted to moderate elevations (800–2000 masl) along the Middle Magdalena River Basin and has only been documented from a few municipalities of the Santander, Boyacá, and northern Cundinamarca departments [18,22]. Adults range from 35 to 70 cm in total length and have a tricolored body pattern with black rings arranged in 17–22 triads [18].

More studies of *Micrurus* venoms in Colombia are needed to identify differences in their composition, clarify potential clinical outcomes that have not been previously considered, and broaden our understanding of individual components and their biotechnological potential. The aim of this study was to characterize the venoms of the Colombian endemic coral snakes *M. medemi* and *M. sangilensis*, as well as those of *M. helleri* populations in Colombia.

## 2. Results

### 2.1. Electrophoresis and Chromatography Analysis

SDS-PAGE showed that the venoms of *M. helleri, M. medemi,* and *M. sangilensis* are composed of protein bands ranging between ~10 and 130 kDa and clustered into two distinct groups. The first one comprises low- and medium-molecular-weight components of approximately 10–15 kDa, and all three venoms show bands around 10 kDa, consistent with the reported mass of three-finger toxins [23]. Bands suggestive of phospholipase A_2_ (PLA_2_) [24] were observed around 15 kDa. The high diversity of bands between 14 and 16 kDa in the venom of *M. medemi* might correspond to C-type lectins (CTL) proteins [25].

The second group of components (>20 kDa) encompasses bands with apparent molecular weights corresponding to metalloproteinases (SVMP; between 20 and 100 kDa) and serine proteases (SVSP; 26 to 67 kDa) [26,27], which appear to be especially abundant in the venom of *M. medemi* (around 40 kDa). All three venoms show evident bands around 55 kDa, and two bands were observed around 70–90 kDa in the venoms of *M. helleri* and *M. medemi*. A conspicuous band near 130 kDa was evident only in the venom of *M. helleri*. These heavy components may account for L-amino acid oxidases (LAAO) and hyaluronidases (HYA) proteins [28,29] (Figure 1).

The chromatographic run revealed 36 fractions covering the entire range of the three venoms. Figure 2 illustrates the composition of the venoms using C18 columns and reveals a high abundance of hydrophobic fractions beyond 55 min in *M. helleri* and *M. sangilensis* (Table 1). These hydrophobic fractions may be associated with serine proteases, C-type lectins, metalloproteinases, and acid L-amino oxidases; this is in contrast to other *Micrurus* venoms in which polar and small/medium-sized components are more abundant [11,30,31,32,33]. For analytical purposes, the chromatogram was segmented for analysis into three sections: (a) from 0 to 38 min, (b) 38 to 50 min, and (c) above 50 min. Based on the small amount of venom available, only the most abundant fraction of each section was subjected to SDS-PAGE. The first and second sections exhibited bands with molecular weights corresponding to 3FTx in all venoms, and the third section was composed of numerous bands ranging from 11 to 16 kDa, especially that of *M. helleri* and *M. sangilensis,* which may correspond to either PLA_2_ or CTL isoforms.

### 2.2. Protein Identification and Quantification

Analysis of the tryptic peptides revealed 92, 93, and 112 total proteins in *M. helleri, M. medemi,* and *M. sangilensis* venoms, respectively. These proteins were further classified as cellular-related components, which are unlikely to be involved in envenomation pathophysiology, and the toxin-related components that accounted for 57.6% (*n* = 53), 64.5% (*n* = 60), and 51.9% (*n* = 58) of the identified proteins in the venoms of *M. helleri*, *M. medemi*, and *M. sangilensis*, respectively, were subsequently grouped into 13 protein families (Figure 3).

Phospholipase components, which include PLA_2_ and phospholipase B (PLB), were the most abundant toxin group in all three venoms, comprising over 35% of the identified proteins. This finding is consistent with other South American *Micrurus* venoms [11,34,35,36]. Three-finger toxins accounted for 14.1 to 17.7% of the identified proteins, followed by snake venom metalloproteinases (9.6–13.8%) (Figure 3).

Snake venom proteinases comprise a minor proportion (less than 5%) of the total toxins in most *Micrurus* venoms reported to date (Appendix A). However, this toxin family comprises nearly 6%, 7.6%, and 12% of the venoms of *M. mipartitus* [37], *M. yatesi* [38], and *M. pyrrhocryptus* [39], respectively. Our results indicate that these proteinases are more abundant in the venoms of *M. medemi, M. sangilensis*, and *M. helleri* (9.6, 11.8, and 13.8%, respectively).

Twenty-two toxin-related proteins were shared among the three *Micrurus* venoms. A pairwise comparison revealed that *M. medemi* and *M. sangilensis* shared 25 toxins, and *M. helleri* shared only 3 toxins with *M. medemi* and another 3 with *M. sangilensis*. This suggests that the venom of *M. helleri* is toxinologically distinct from that of the other two species (Figure 4). The greater toxinological similarity between *M. medemi* and *M. sangilensis*, which are separated by the Andes, than between *M. helleri* and *M. medemi*, which have overlapping geographic distributions, is noteworthy.

Another interesting finding is the unique presence of vespryn in *M. sangilensis* venom. Vespryn has been previously reported to be an infrequent toxin in Brazilian and Costa Rican *Micrurus* species (*M. alleni*, *M. lemniscatus, M. carvalhoi, M. spixii,* and *M. nigrocinctus*) [40], suggesting that an expanded venomic sample can improve our understanding of the biochemical composition of the venom of phylogenetically and geographically distant *Micrurus* species. In addition, hyaluronidases are 4.5 and 3 times more abundant in *M. helleri* than in *M. medemi* and *M. sangilensis*, respectively. In contrast, C-type lectins (CTL) are 13.5 and 10.8 times more abundant in *M. medemi* and *M. sangilensis* than in *M. helleri,* respectively. Serine proteases (SVSP) are almost 13 times more abundant in *M. helleri* and *M. sangilensis* than in *M. medemi.* These findings are shown in Figure 5.

### 2.3. Enzymatic Activities

All *Micrurus* venoms exhibited PLA_2_ activity using both methodological approaches. However, in the egg yolk solution preparations, *M. helleri* and *M. sangilensis* showed higher activity, as indicated by the larger diameter of the translucent halo (an average of 107 mm) in *M. helleri* and *M. sangilensis* than in *M. medemi* (79 mm). *M. helleri* venom exhibited the highest activity according to the colorimetric method, indicating that the optical density of *M. helleri* venom is higher than that of *M. medemi* and *M. sangilensis.* In both methodological approaches, all *Micrurus* venoms had higher PLA_2_ activity than the positive control (*Crotalus durissus cumanensis* venom) (Figure 6a,b).

Hyaluronidase activity was detected in all *Micrurus* venoms. *M. helleri* exhibited a prominent band of 75.9 kDa, *M. medemi* exhibited a prominent band of 70.3 kDa, and *M. sangilensis* exhibited two bands of 72.4 and 67.9 kDa (Figure 6c). Similar results were observed in protease activity. The activity of *M. helleri* venom was 1.6 and 1.9 times higher than that of *M. medemi* and *M. sangilensis*, respectively. These results could be explained by the presence of protein families such as serine and metalloproteinases in the proteome of all three species, which also showed higher activity in *M. helleri* venom than in the other two species (Figure 6d).

### 2.4. Venom Toxicity

*M. medemi* venom exhibited the highest lethality (8.79 µg/mice, CI95% = 7.16–10.80), followed by the venom of *M. sangilensis* (15.85 µg/mice, CI95% = 11.5–21.7) and *M. helleri* (22.87 µg/mice, CI95% = 17.79–29.4) (see Table 2).

Cytotoxicity was observed in all venoms and was assessed using three different approaches. *M. helleri*, *M. medemi*, and *M. sangilensis* venoms displayed a significant decrease in cell viability in hippocampal neuron cultures, with notable differences observed among species (ANOVA = 14.75, *p*-value < 0.001). *M. helleri* venom led to the lowest cell viability (≤20%) with a venom concentration of less than 0.1 μg/mL, and low cell viability was observed across the range of concentrations tested. Conversely, *M. medemi* and *M. sangilensis* venoms demonstrated dose-response behavior, with a significant decrease in cell viability between concentrations of 50 and 100 µg/mL, resulting in viabilities of up to 35% and 18%, respectively (Figure 7a). Similarly, in the HTB-132 cell line assays, all venoms exhibited a dose–response pattern, with a significant decrease in cell viability between concentrations of 50 and 100 µg/mL, resulting in viabilities of up to 18%, 25%, and 18% for the venoms of *M. helleri* (median inhibitory concentration—IC_50_ = 3.5 µg/mL)*, M. medemi* (IC_50_ = 24.5 µg/mL), and *M. sangilensis* (IC_50_ = 14.5 µg/mL), respectively (Figure 7b). In PC3 cell line assays, *M. helleri* and *M. medemi* venoms resulted in a significant decrease in cell viability. *M. helleri* venom exhibited a dose–response pattern between concentrations of 6 and 100 µg/mL, which resulted in a decrease in cell viability of up to 15% (IC_50_ = 25.8 µg/mL). *M. medemi* exhibited a dose–response pattern between concentrations of 25 and 50 µg/mL, which led to a decrease in cell viability by up to 20% (Figure 7c).

## 3. Discussion

This work presents the first biochemical and biological characterization of the venoms of *M. medemi* and *M. sangilensis*, two endemic and poorly known Colombian coral snake species. The venom of a subpopulation of *M. helleri*, which is a taxonomically controversial species, was also characterized. Our samples of *M. helleri, M. medemi,* and *M. sangilensis* comprised several members of the PLA_2_ and 3FTx families, which is consistent with the results of studies of other South American coral snake venoms [16,17,28,35,36,37,40,41,42], according to electrophoretic, chromatographic, and proteomic analysis. The properties of *Micrurus* venoms vary greatly among species, which is attributed to their diverse habitats and prey preferences as well as phylogenetic inertia [12,43,44,45,46]. 

*Micrurus* venoms are composed of approximately 19 different toxin families, and the PLA_2_ and 3FTx families are the main components [12,42]. Our results suggest that electrophoresis bands with low molecular masses (between 6 and 10 kDa) were observed in *M. helleri*, *M. medemi,* and *M. sangilensis* venoms, indicating the presence of 3FTXs (Figure 1). These same electrophoresis bands have been previously reported as venom fractions corresponding to the same neurotoxins in several *Micrurus* venoms [31,47,48,49,50]. Likewise, electrophoresis bands ranging from 13 to 15 kDa and observed in all the venoms assessed indicate the presence of PLA_2_ proteins, which is consistent with the electrophoretic migration previously reported for the venoms of *Micrurus* species [11,42,47,51,52].

However, the wide range of electrophoresis bands observed among the *Micrurus* species suggests a highly variable ratio of 3FTxs, which is associated with the diverse array of toxin components. Although each species exhibits a distinct venom profile, some features are shared with other Andean (e.g., *M. mipartitus*) and Amazonian species (e.g., *M. spixii*) [32,33,45,53,54]. Chromatograms of the venoms of the three *Micrurus* species were consistent with the species-specific venom profiles, as suggested by the electrophoresis bands observed. The venoms had abundant components in the high molecular (and hydrophobic) range, which suggests the possible presence of L-amino acid oxidases (Figure 2) [33]. This finding is consistent with the literature on the clinical manifestation of snakebite envenomation caused by *M. helleri*, which includes thrombocytopenia and mild coagulopathy, activities that have been tentatively associated with the effect of LAAO on platelet aggregation; however, conclusive evidence of these activities is lacking [55,56].

The peaks observed at minutes 50 and 72 in *M. helleri* venom may be related to PLA_2_ but also to metalloproteinases or serine proteases. This speculation is supported by the thrombin-like activity previously reported in vitro and the protease activity observed in this study for the same venom [45,55]. Further research, including thorough characterizations, is needed to establish the identity of these signals. Nevertheless, the chromatographic profiles of *M. helleri, M. medemi*, and *M. sangilensis* venoms primarily exhibited prominent signals in the first 38 min of the retention time, which is consistent with the typical retention times of 3FTx and PLA_2_ toxins reported in *Micrurus* species venoms [12,54,57].

Our chromatographic profile of *M. helleri* venom, as well as the results of a previous proteome analysis conducted on Amazonian populations of the same species [36], supports the molecular phylogenetic hypothesis proposed by Hurtado-Gómez et al. [58], which posits that *M. helleri* comprises two non-sister lineages, one occurring along the Andean foothills and the other in lowland Amazonia (Leticia, Colombia). The lowland Amazon lineage of *M. helleri* lacks the two signals with retention times above 60 min in our chromatographic profile of the Andean foothills lineage. In addition, our proteome profile reveals that the abundances of PLA_2_ (22% less) and 3FTx (7% less) are low compared with those reported for lowland Amazonian populations by Sanz et al. [36]. This is made up for by the presence of metalloproteinases, serine proteases, and LAAO, which are not abundant in *M. helleri* or other *Micrurus* venoms from Colombia [11,33]. Our findings are thus consistent with the taxonomic proposal of Hurtado-Gómez et al. [58] that the name *M. helleri* should be restricted to the populations of the eastern Andean foothills.

In addition, the proteomic results confirm the findings of Lomonte et al. (2016) [34] regarding the existence of two distinct patterns in the proteomic profiles in *Micrurus* venoms; the divergence of venom phenotypes can be of evolutionary significance, and PLA_2_ and 3TFx are the dominant toxin families. Our study revealed that both protein families accounted for more than 55% of the total proteome, indicating that the venoms were PLA_2_-dominant in all cases (Figure 3). Furthermore, the mass spectrometry analysis of tryptic-digested peptides provided additional evidence supporting the patterns observed in the RP-HPLC and electrophoretic profiles. Many compounds related to high-molecular-weight toxins, such as SVMP, LAAO, SVSP, and CTL, were identified. These findings are consistent with the results of previous studies by Olamendi-Portugal et al. (2018) [39], which further indicates that our data are reliable.

The proteomic profiles of coral snake venoms follow a divergent latitudinal pattern in which one group, the long-tailed monadal species (mostly North and Central American) are characterized by the predominance of PLA_2_ (>45–58% of total venom proteins) and a second group, formed by short-tailed, triadal species (mostly South American), show a predominance of 3FTx (>48–95%) [34]. However, the venoms of the short-tailed triadal species *M. helleri*, as well as the two monadal short-tailed species *M. medemi* and *M. sangilensis*, are dominated by phospholipase components with PLA_2_ comprising 40.6%, 43.1%, and 30.4% of the total venom proteins and 3FTx comprising 14.1%, 17.7%, and 17.7% of the total venom proteins, respectively. Similar results have been obtained for *M. helleri* from the Amazonian region of Colombia and Ecuador; in Colombia, PLA_2_ and 3FTx comprise 62.5% and 21.1% of the total venom proteins, respectively, whereas in Ecuador, PLA_2_ and 3FTx comprise 72.1% and 17.8% of the total venom proteins, respectively [36,37]. The proteomic composition of various coral snake species reported over the last four years, including this work, does not support the hypothesis of the divergent latitudinal pattern nor a triadal vs. monadal (or long-tailed vs. short-tailed) divergence for explaining the phospholipase/3FTx-dominant phenotypes (see Appendix A). The geographical distribution of these venom phenotypes, in which northern species exhibit the phospholipase-dominant proteomic profile and South American species have venoms dominated by 3FTx, could be better clarified using more comprehensive population sampling, evolutionary analyses of coral snake species, and analyses of the composition of the venoms of more coral snake species.

The enzymatic activities of the three *Micrurus* venoms varied. PLA_2_ activity was detected in all venoms regardless of the method used, and it was highest in *M. helleri* and *M. sangilensis,* followed by *M. medemi* (Figure 6). The variability observed among species, as well as the high reactivity of *M. helleri* venom, was expected and consistent with that reported by Tanaka [44] and Aird [59].

Furthermore, hyaluronidase activity in all *Micrurus* venoms was confirmed, as evidenced by bands around 67.9–75.9 kDa, suggesting that there is substantial dispersion in situ; hyaluronidase activity is based on the degradation of hyaluronic acid, which is present in the extracellular matrix and acts as a spreading factor [60]. Approximately 5% of the proteome of *M. helleri* comprised hyaluronidases, which was indicated by the marked degradation bands in the zymogram. Hyaluronidase activity was observed in the venoms of some Colombian species, such as *M. dumerilii*, which had bands around 65 kDa [12]; these bands have also been commonly observed in coral snake species in Central and South America [11,61]. However, hyaluronidase activity is a common feature of *Micrurus* venoms and has been widely observed in coral snake species throughout Central and South America [11,61].

Protease activity was observed in all venoms and significantly differed between *M. helleri* and the other two species. This might stem from the higher number of serine and metalloproteinase toxins detected by proteomic analysis. To the best of our knowledge, no previous study of *Micrurus* venoms has reported such a finding.

High levels of serine and metalloproteinase proteins in the venoms (10% for *M. medemi* and 17.6% for *M. sangilensis*) also suggest enzymatic activity or poor substrate recognition. Metalloproteinases are common toxins in viperid species but rare in elapid species. Some elapid species such as *Naja naja* (common cobra) and *Naja melanoleuca* (black and white cobra) have significant amounts of metalloproteinases in their proteomes [54]. Nevertheless, the metalloproteinases in *Micrurus* venoms are minor components of the proteome, comprising less than 5% of the total venom, and serine proteases, if detectable, comprise less than 2% of the venom components; thus, their enzymatic activity tends to be minimal [16,37,53,55]. Low-activity protease-like enzymes have been identified in *M. dumerilii* and *M. alleni* venoms, and the percentages of these enzymes in the proteome are low [11,61]. These enzymes can be found between 25 and 68 kDa, and serine proteases between 15 and 30 kDa. Thus, the bands observed in the SDS-PAGE for all venoms could be related to these family groups [44,62].

The most relevant clinical manifestations elicited by the venom of *Micrurus* species are those related to neuromuscular impairment, and coagulopathies have not been reported [63,64]. Nevertheless, envenomation by *M. lemniscatus helleri* [65] induces a slight prolongation in prothrombin (PT) and partial thromboplastin time (PTT) along with mild thrombocytopenia (from 83 to 41 × 10^9^ cells/L). Envenomation by *M. annellatus* causes the blood to be incoagulable for two days after the bite even after the administration of antivenom; slight thrombocytopenia (from 164 to 133 × 10^3^ cell/mL) has also been observed [66]. Coagulation problems and thrombocytopenia are caused by some metalloproteinases [67] and serine protease members [68]. Our results suggest that the biological and clinical role of these proteinases in the envenomation process produced by South American coral snakes merits further attention.

Among the Colombian *Micrurus* venoms characterized, the lethality of *M. helleri, M. medemi*, and *M. sangilensis* venoms can be considered moderate to high (Table 1). The venom of *M. medemi* ranks second among Colombia’s coral snake species in terms of lethality, surpassed only by *M. isozonus* [41]. *M. sangilensis* and *M. helleri* rank fourth and fifth, respectively, indicating that their toxicity is moderate compared with the known lethality range of Colombian *Micrurus* species [41]. The LD_50_ (0.9 μg/g mice) of *M. sangilensis* exceeds that of *M. surinamensis* (2.15–4.35 μg/g mice) [44] and is near that of *M. dumerilii* (0.8–1.9 μg/g mice) [11], the former being rich in 3FTx and the latter rich in PLA_2_ [28,56]. However, the LD_50_ of *M. medemi* venom (0.5 μg/g mice) is similar to the LD_50_ values of *M. mipartitus*, *M. clarki*, and *M. nigrocinctus* venoms [31].

These findings have important implications for snakebite accidents as a public health issue. First, the moderate to high toxicity of the venoms of *M. medemi* and *M. sangilensis* are of particular concern because these snakes inhabit urban and peri-urban areas. Snakebites caused by these species thus pose high potential risks and constitute a medical emergency [69]. Second, among the antivenom therapies currently available in Colombia capable of treating elapid envenomation, none include any of the three venoms as immunogens, but their neutralizing effectiveness against *M. helleri* and *M. medemi* has been reported based on their cross-reactivity, such as for the anticoral antivenom produced by the Colombian Health National Institute (INS Spanish acronym) [41] and Instituto Butantan [44,70].

The differences observed in the hippocampal neuronal assays between the consistently low cell viability, observed across the range of concentrations tested for *M. helleri* venom, and the dose-dependent responses, observed for *M. medemi* and *M. sangilensis* venoms, (Figure 7a) suggest that *M. helleri* venom possesses potent neurotoxic effects and exhibits a specific affinity for nervous tissue. This is consistent with the findings of Oliveira et al. [71] showing that the injection of various concentrations of the Mlx-8, Mlx-9, Mlx-10, and Mlx-11 fractions of *M. lemniscatus* (*sensu lato*) venom in rats did not result in mortality. However, when the venom was administered through the intrahippocampal route, envenomation and death were observed. These results indicate that certain *Micrurus* species have toxins that specifically target hippocampal tissue rather than produce a generalized effect on the organism. In hippocampal cells, the effect of *M. helleri* venom was replicated by *M. sangilensis* venom at a dose of 100 μg/mL.

All three venoms exhibited a dose-dependent response with a less pronounced reduction in cell viability on HTB-132 than that observed in the hippocampal neuronal culture, which provides further evidence of their specific affinity for nervous tissue rather than a generalized effect. The *M. helleri* and *M. sangilensis* venoms exhibited a similar pattern of reducing cell viability in the HTB-132 cell line. Notably, the ~25 μg/mL concentration of *M. medemi* venom resulted in a significant reduction in viability to approximately 25% in the HTB-132 line.

The MTT assay could only be conducted on *M. helleri* and *M. medemi* venoms for the PC3 cell line because the amount of *M. sangilensis* venom available was insufficient. *M. helleri* venom reduced the cell viability of the PC3 line to below 20% at high concentrations. Although the venom of *M. sangilensis* could not be tested on this cell line, the trend suggests that the venom of *M. helleri* contains the most potent cytotoxic components. Previous studies have described the cytotoxicity of elapid venoms on tumor cells, which is attributed to toxins such as disintegrins [72], cytotoxins of the 3FTxs family [73], and L-amino acid oxidases (LAAO) [74]. Additionally, PLA_2_ has been shown to affect the central nervous system [75,76]. Both the *M. helleri* and *M. medemi* venoms displayed a dose-dependent response and significantly reduced cell viability. These findings will aid future research on venom-specific toxins, including studies of their antitumor activity against breast and prostate cancer cells.

## 4. Conclusions

Our findings enhance our understanding of the 3FTx/PLA_2_ dichotomy proposed by several authors regarding the relative abundance of these toxins in *Micrurus* venoms. These preliminary trials shed light on understudied coral snake venoms. The protease activity in the *Micrurus* venoms examined is lower than that in viper venoms; however, significant hyaluronidase activity, a key factor in venom dispersion, was observed. These venoms exhibited cytotoxic activity against tumor cells, which makes them potential candidates for identifying targets involved in cell lysis events, interactions with ion channels, and the modulation of inflammatory mediators. These characteristics make them valuable for exploring their potential applications in the management of chronic diseases, such as cancer and infections caused by microorganisms.

## 5. Materials and Methods

### 5.1. Target Species and Venoms

Lyophilized pooled venoms of *M. medemi* (Villavicencio, Meta), *M. sangilensis* (San Gil, Santander), and *M. helleri* (Villagarzón, Putumayo) were used; *Bothrops asper* and *Crotalus durissus cumanensis* venoms were used as controls. Venoms were provided by the venom bank of the Instituto Nacional de Salud. They were stored below –20 °C for preservation following standards established by the World Health Organization (WHO) and INS protocols [77,78].

*M. helleri* has been a controversial taxon. Originally, this taxon was considered a subspecies of *M. lemniscatus*, distributed across the Amazon region to the foothills of the Andes from northern Brazil, southern Venezuela, and Colombia to Ecuador, Peru, and Bolivia [18,79]. However, a molecular phylogenetic analysis conducted by Hurtado-Gómez et al. [58] has shown that *Micrurus lemniscatus* (*sensu stricto*) does not occur in Colombia. Although Colombian populations of the Amazon region and eastern foothills of the Cordillera Oriental of Colombia were traditionally assigned to *M. lemniscatus,* these populations represent two non-sister lineages, one occurring along the Andean foothills and the other in lowland Amazonia. According to Hurtado-Gómez et al. [58], *M. helleri* is restricted to the populations of the Andean eastern foothills from Peru to Colombia. We adopted this taxonomic classification given that the pooled venoms used in our assays were derived from Amazonian foothill localities (Villagarzón, Putumayo department).

### 5.2. Experimental Animals

The lethal dose test was performed on *Mus musculus* ICR CD–1 mice of both sexes with weights between 16 and 20 g supplied by animal facilities at the INS. During the experimental trial, the animals were not subjected to additional stress aside from venom inoculation. Mice were fed ad libitum during the experiments. The procedures used during the experiments were approved by the Animal Ethical Use Committees of the Instituto Nacional de Salud de Colombia (protocol INT–R04.0000–001), as well as the Ethical Research Committee of the Medicine School of the Universidad Nacional de Colombia under the act 007–095–18. This study was conducted under the protocol of the Colombian animal welfare law (Ley 1774, 2016) and the Universal Declaration on Animal Welfare (UDAW) endorsed by Colombia in 2007.

### 5.3. Venom Characterization

#### 5.3.1. SDS-PAGE

Venom was reconstituted, and protein was quantified by dissolving 5 mg of each venom in 5 mL of phosphate-buffered saline (PBS). The protein concentration was determined by the bicinchoninic acid (BCA) method, using 1 mg/mL bovine serum albumin (BSA) as a standard [80]. Venom mixtures were centrifuged at 5000 rpm for five minutes, and the supernatant was collected for subsequent analysis.

Sodium dodecyl sulfate–polyacrylamide gel electrophoresis (SDS-PAGE) was performed following the procedures described by Laemmli in 15% gels [81,82]. Gels were stained with Coomassie Blue G–250 solution and/or silver stain [83,84]. Gels were analyzed using Bio–Rad ImageLab 6.1 software.

#### 5.3.2. Reversed-Phase Chromatography (RP-HPLC)

Briefly, 2 mg of venom was dissolved in 30 μL of solution A (water, containing 0.1% trifluoroacetic acid—TFA) and separated on a C18 column (Discovery^®^, 5 μm particle diameter; 250 × 4.6 mm) using a Hitachi LaChrom Elite chromatograph monitored at 215 nm. Elution was performed at a flow rate of 1 mL/min with the following gradient of solution B (acetonitrile, containing 0.1% TFA): 0% B for 5 min, 0–15% B over 10 min, 15–45% B over 60 min, 45–70% B over 10 min, holding at 70% for 10 min, and 70–100% B over 13 min. This procedure was based on that described by Rey-Suarez with modifications [33,52,85].

### 5.4. Protein Identification and Quantification

#### 5.4.1. Protein Digestion with Trypsin

Venom from each species was subjected to trypsin digestion prior to identification by mass spectrometry. Protein extracts were diluted in a 6 M urea buffer and reduced by adding 2.5 µL of the reduction buffer (45 mM DTT, 100 mM ammonium bicarbonate) for 30 min at 37 °C; this was followed by alkylation through the addition of 2.5 µL of the alkylation buffer (100 mM iodoacetamide, 100 mM ammonium bicarbonate) for 20 min at 24 °C in the dark. Prior to digestion, samples were diluted with water to reduce the urea concentration to 2 M. A protein/trypsin ratio of 20 was used for the tryptic digestion (trypsin sequencing grade from Promega, 50 mM ammonium bicarbonate). Protein digestion was performed at 37 °C for 18 h and stopped with 5 µL of 5% formic acid. Protein digests were dried down in a vacuum centrifuge and stored at −20 °C until LC-MS/MS analysis.

Scaffold (version Scaffold_4.8, Proteome Software Inc., Portland, OR, USA) was used to validate the MS/MS-based peptide and protein identifications. Peptide identifications were accepted if they could be established at greater than 95.0% probability by the Peptide Prophet algorithm [86] using the Scaffold delta-mass correction. Protein identifications were accepted if they could be established at greater than 95.0% probability. Protein probabilities were assigned by the Protein Prophet algorithm [87] with an FDR < 0.1% and minimum number of peptides of 2.

#### 5.4.2. LC-MS/MS

Before LC-MS/MS, the protein was re-solubilized under agitation for 15 min in 10 µL of 0.2% formic acid. Desalting/cleanup of the digests was performed using C18 ZipTip pipette tips (Millipore, Billerica, MA). Eluates were dried with a Speed-vac, reconstituted under agitation for 15 min in 12 µL of 2%ACN-1%FA, and loaded into a 75 μm × 150 mm, Self-Pack C18 column in the Easy-nLC II system (Proxeon Biosystems). Peptides were loaded on a column and eluted with a three-slope gradient at a flow rate of 270 nL/min. Solvent B first increased from 1 to 33% in 110 min, from 33 to 65% B in 10 min, and then from 65 to 84% B in 2 min. The HPLC system was coupled to an Orbitrap Fusion mass spectrometer (Thermo Scientific, Waltham, MO, USA) through a Nanospray Flex Ion Source. Nanospray and S-lens voltages were set to 1.3–1.8 kV and 50 V, respectively. The capillary temperature was set to 250 °C. Full-scan MS survey spectra (m/z 300–1500) in profile mode were acquired in the Orbitrap with a resolution of 120,000 with a target value at 3 × 10^5^. The most intense peptide ions were fragmented by HCD (charges 2–4) and/or ETD (charges 4–7) and analyzed in the Orbitrap at a resolution of 15,000, and an AGC target value set to 5.0 × 10^4^. The peptide ion fragmentation parameters were as follows: reaction time of 120 ms, reagent target of 2.0 × 10^5^, a maximum reagent injection time of 200 ms for ETD, and a normalized collision energy of 30% for HCD. Target ions selected for fragmentation were dynamically excluded for 45 s after 3 MS/MS scan events.

#### 5.4.3. Protein Identification

The peak list files were generated with Proteome Discoverer (version 2.4) using the following parameters: minimum mass, 500 Da; maximum mass, 6000 Da; no grouping of MS/MS spectra; precursor charge, auto; and minimum number of fragment ions, 5. Protein database searches were performed using Mascot 2.6 (Matrix Science) against the UniProt Serpentes database. The mass tolerances for precursor and fragment ions were set to 10 ppm and 0.6 Da, respectively. Cleavage was performed using trypsin, and up to 1 missed cleavage site was allowed. Methionine oxidation and carboxylation of glutamic acid were specified as variable modifications. Data interpretation was performed using Scaffold (version 4.8).

#### 5.4.4. Protein Quantification

The NSAF was used to obtain rough protein abundance estimates for each protein in all venoms. A Python script was used to sort and structure the total spectral count data for subsequent quantitative analysis of proteins using the normalized spectral abundance factor (NSAF) according to Zybailov et al. (2006) [88]. NSAF was used because large proteins are known to contribute more to peptides and spectral counts; NSAF considers the number of spectral counts (Sc) from a protein divided by the number of amino acids (L), and this value is finally divided by the sum of Sc/L from all identified proteins.

### 5.5. Biological Activities

#### 5.5.1. Phospholipase A_2_ activity

Two approaches were used to assess the PLA_2_ activity. First, a 10% (*w/v*) egg yolk solution in agarose was made; once the solution solidified, 5 µg of the venom of each *Micrurus* species was seeded into circular wells in solution, solidified, and incubated for 1 h at 37 °C [60]. *Crotalus durissus cumanensis* venom from the Magdalena Medio ecoregion was used as a positive control, and phosphate saline buffer (pH 7.4) was used as a negative control. Both control venoms were dissolved and seeded in egg yolk solutions in the same manner as described for the *Micrurus* species venoms under study. Finally, phospholipase activity was visualized in a UV transilluminator through the formation of translucent or cloudy halos.

The second approach followed the colorimetric method [89,90]. A total of 0.6 g of lecithin was dissolved in 1 mL of ethanol at 45 °C, followed by 0.86 mL of Triton X–100, 8 mL of NaCl (0.1 M), 4 mL of phenol red (5.5 mm), 1.6 mL of CaCl_2_ (1 M), and 16.6 mL of distilled H_2_O. The pH of the solution was adjusted by adding a drop of 2 M NaOH. Different concentrations of protein ranging from 0.0005 to 0.6600 μg/μL were seeded in 96–well culture plates, and 100 µL of the previously described substrate was added to each well. The plates were incubated for 15 min at 37 °C. Deionized water was used as a negative control, and *C. durissus cumanensis* venom was used as a positive control. The optical density was read at 540 nm on a Bio–Rad Model 550 Microplate Reader. The assays were conducted in triplicate, each with three technical replicates.

#### 5.5.2. Hyaluronidase Activity

Five μg of the crude venoms of *M. helleri*, *M. medemi,* and *M. sangilensis* were dissolved in 1X PBS. Samples were run on 10% SDS-PAGE gel and co-polymerized with 0.5 mg/mL hyaluronic acid (HA) (hyaluronic acid sodium salt from *Streptoccocus equi*, Sigma^®^, St. Louis, MI, USA). A volume of 4 μL of unheated Laemmli sample buffer (under non-denaturing conditions) was added. After electrophoretic separation, the gels were incubated with PBS (pH 5.8) (0.1 M phosphate buffer, 0.15 M NaCl, and 5% Triton X–100) for one hour at room temperature. This procedure was repeated twice. Afterwards, the samples were incubated with PBS (pH 5.8) (0.1 M phosphate buffer, 0.15 M NaCl, and 5% Triton X–100) for 1 h and then in PBS (pH 5.8) (0.1 M phosphate buffers and 0.15 M NaCl) for 10 min. Subsequently, the gels were run in a wet chamber overnight. Two washes were performed with Tris–HCl (pH 7.95, 0.015 M) and stained for 5 h with agitation using a 5% formamide solution, 20% isopropanol, Tris–HCl (pH 7.95, 0.015 M), and 5 mL of 0.1% Stains-All, protected from light exposure. The gels were destained for 1 h under constant agitation with 5% formamide, 20% isopropanol, and Tris–HCl (pH 7.95, 0.015 M). Hyaluronidase activity was determined as being in the gel region that was not stained [91]. The gels were analyzed using Bio–Rad ImageLab 6.0 software.

#### 5.5.3. Protease Activity

Based on the supplier’s protocol, a total volume of 100 μL was required, including 50 μL from the sample or control (diluted in digestion buffer) and the other 50 μL from the substrate containing the fluorophores (BODIPY mixture). For the assay, 3 μg of each sample was added, the digestion buffer was added until a 50 μL volume was achieved, and 50 μL of the BODIPY mixture was added to a black 96-well microplate. The plate was protected from light, and the fluorescence was determined using a Synergy HT^®^ microplate reader with excitation at 480 nm and fluorescence emission at 520 nm. To monitor the kinetics, readings were taken every 2 min for one hour.

#### 5.5.4. Lethal Dose

Serial dilutions of venom in 500 µL of 0.85% NaCl were inoculated via the intraperitoneal route into groups of five mice. Evaluations were conducted using seven to nine dilutions of the venom of each snake species with a dilution factor between 1.5 and 1.7 and a concentration range of 0.800 to 68.344 µg/mouse. A total of 500 µL of 0.85% NaCl was inoculated in the same manner in the negative control. The survival times of the experimental animals were determined to identify the dose of venom (µg/mice) that causes 50% of the population’s death (LD_50_). The death ratio was read after 48 h, and the experiments were considered valid only when they reached either zero or 100%. LD_50_ was expressed in micrograms of venom (μg) per mice. 

#### 5.5.5. Cytotoxicity

Three approaches were used to assess the cytotoxicity of the *M. helleri*, *M. medemi,* and *M. sangilensis* venoms. First, cytotoxicity was evaluated in primary cultures of hippocampal neurons from 18-day-old fetuses of Wistar rats. This procedure was performed in primary neuron growth medium (PNGM) (composed of 2% of Primary Neuron Basal Medium (PNBM) NSF–1, 2 mM L–Glutamine, gentamicin 50 µg/mL, and 37 ng/mL of amphotericin) in 96-well plates at a density of 20,000 cells/well in 100 µL of PNGM. This preparation was incubated at 37 °C in a wet atmosphere and 5% CO_2_. After six days of in vitro neuronal differentiation, *M. helleri*, *M. medemi,* and *M. sangilensis* venoms were added in a range of concentrations between 1 and 100 µg/mL; the PNGM medium was used as a negative control to determine 100% cell viability for 24 h.

Second, cytotoxicity was evaluated in the HTB-132 cell line (adenocarcinoma derived from mammary glands, pleural effusion of the metastatic site, with adherent epithelial cells), and cells were seeded in 96-well plates at a density of 10,000 cells/well in 100 µL of RPMI culture medium to observe dose-response effects on cell viability. These cells were treated for 36 h at 37 °C with 5% CO_2_ and 95% humidity, with each of the venoms in a range of concentrations between 1 and 100 µg/mL. Finally, the cytotoxicity of *M. helleri* and *M. medemi* venoms was assessed in the PC3 line cell (type IV adenocarcinoma prostate cancer derived from bone metastasis with adherent epithelial cells).

Cytotoxicity activity was assessed using the MTT colorimetric assay [80]. At the end of the assay, 10 μL of MTT was added (5 mg/mL in 1X PBS) to each well and incubated for 3 h. Subsequently, the culture medium was discarded, and the formazan crystals were air-dried and solubilized with 100 μL of 100% DMSO for 30 min at 37 °C. Once the solubilization process was completed, the absorbance was measured at 540 nm [80]. Each trial was performed in triplicate.

### 5.6. Statistical Analysis

LD_50_ and respective 95% confidence intervals were established using the Spearman–Kärber method [92,93,94]. To compare the cytotoxicity activities of *M. helleri*, *M. medemi*, and *M. sangilensis* venoms, a one-factor ANOVA was performed, followed by Tukey’s test for multiple comparisons [95].

## Figures and Tables

**Figure 1 toxins-15-00622-f001:**
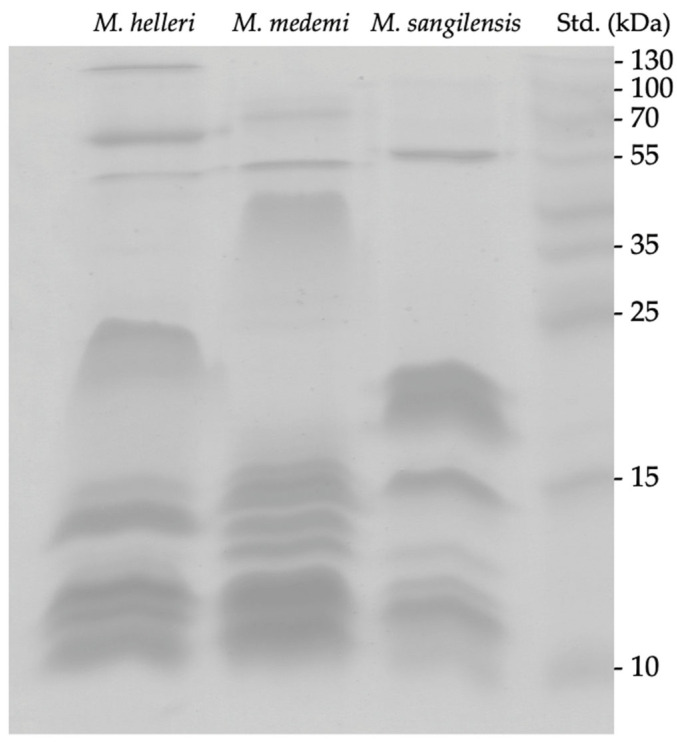
Separation of *M. helleri, M. medemi,* and *M. sangilensis* venoms obtained using 15% SDS-PAGE under reducing conditions. Each lane was seeded with 20 μg of protein. Std.: molecular weight standard.

**Figure 2 toxins-15-00622-f002:**
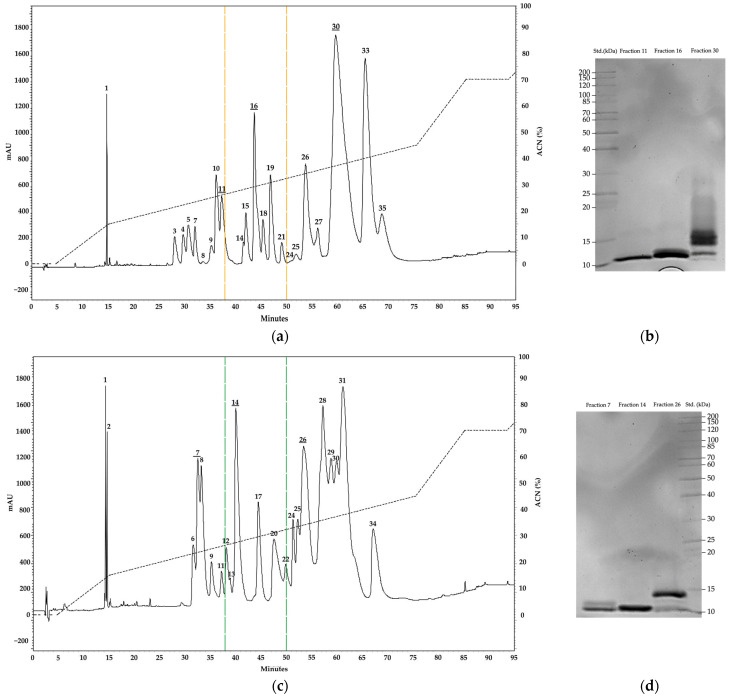
Chromatographic profiles of whole venoms on a C18 column (Discovery^®^, 5 μm particle diameter; 250 × 4.6 mm) highlighting more abundant fractions (bold and underlined) subsequently observed in 12.5% SDS-PAGE under reducing conditions. *M. helleri* (**a**,**b**), *M. medemi* (**c**,**d**), and *M. sangilensis* (**e**,**f**). For analytical purposes, the chromatograms were divided into three sections, the limits of which are indicated by dotted lines. See the text for more detail. Std.: molecular weight standard (kDa).

**Figure 3 toxins-15-00622-f003:**
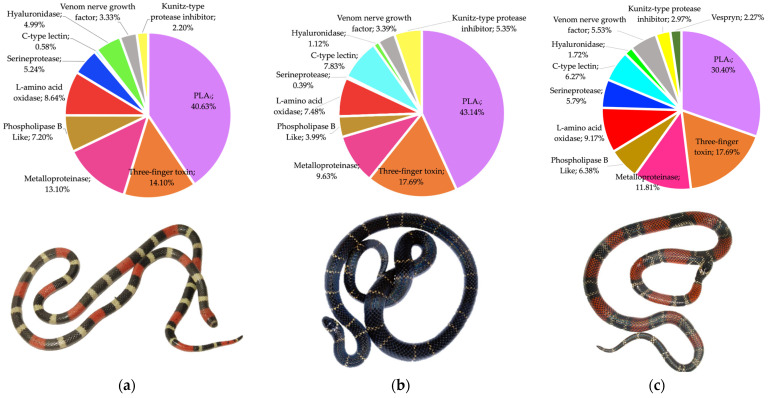
Percentages of the protein families in the venoms of (**a**) *M. helleri*, (**b**) *M. medemi,* and (**c**) *M. sangilensis*. Snake photos provided by Juan Pablo Hurtado-Gómez.

**Figure 4 toxins-15-00622-f004:**
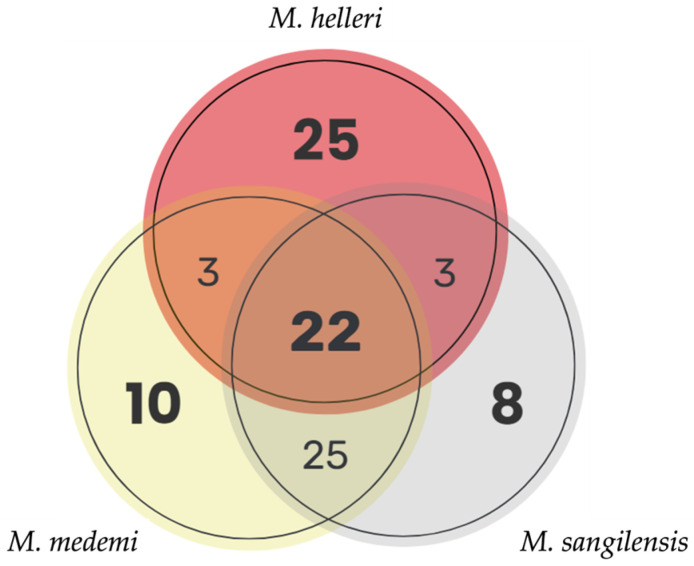
Comparison of the number of shared and unique toxin-related proteins among the venoms of *M. helleri* (red; *n* = 53), *M. medemi* (yellow; *n* = 60), and *M. sangilensis* (light grey; *n* = 58). Note the high number of shared proteins between *M. medemi* and *M. sangilensis* and the higher number of unique peptides in *M. helleri* compared with *M. medemi* and *M. sangilensis*.

**Figure 5 toxins-15-00622-f005:**
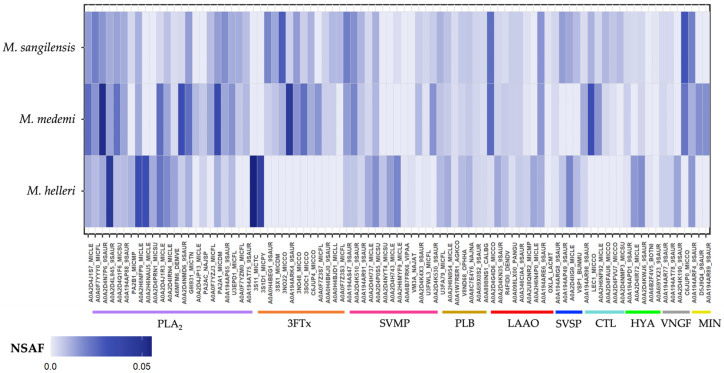
Relative abundances of identified proteins using the normalized spectral abundance factor (NSAF). The color represents the protein families in this study. PLA_2_: phospholipases A_2_, 3FTx: three-finger toxins, SVMP: snake venom metalloproteinases, PLB: phospholipase B, LAAO: L-amino acid oxidases, SVSP: snake venom serine proteases, CTL: C-type lectin, HYA: hyaluronidases, VNGF: venom nerve growth factor, and MIN: minority compounds.

**Figure 6 toxins-15-00622-f006:**
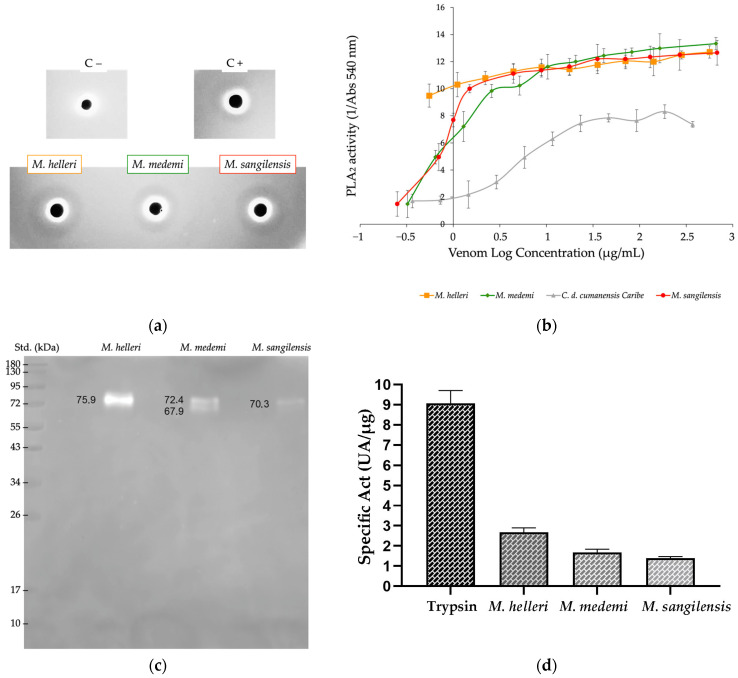
Enzymatic activities for *M. helleri, M. medemi,* and *M. sangilensis* venoms. (**a**) Phospholipase A_2_ assay in agarose and 10% egg yolk solution using 5 μg of each venom. The translucent halos formed around each well can be observed. Top: phosphate buffered saline (pH 7.4) used as the negative control (C−), and *Crotalus durissus cumanensis* venom used as the positive control (C+) (5 μg). Bottom: *M. helleri*, *M. medemi,* and *M. sangilensis*. (**b**) Determination of phospholipase A_2_ activity by a colorimetric assay in a medium containing lecithin as the substrate (triplicate). Bars denote ± standard error. Panel (**c**) shows hyaluronidase activity. Non-stained areas depict positive activity. Std.: molecular weight standard. (**d**) Protease activity of the venoms obtained using an EnzChek^®^ Protease Assay Kit. Trypsin was used as a positive control. Values represent the mean of three replicates. Bars denote standard error.

**Figure 7 toxins-15-00622-f007:**
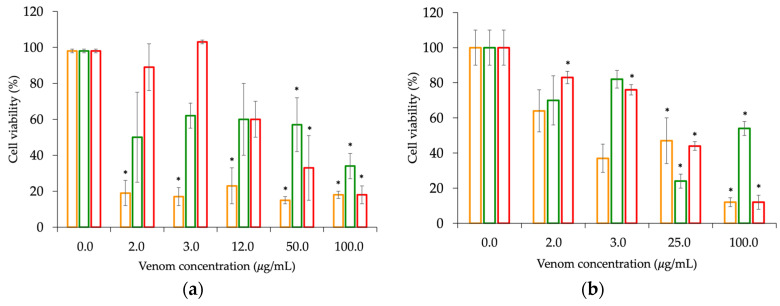
Cell viability of three cell lines due to the effect of *Micrurus* venoms. (**a**) Hippocampal neuronal assay. Bar plots illustrate the percentage of cell viability in neuronal primary cultures after exposure to varying concentrations (2, 3, 12, 50, and 100 μg/mL) of the venoms. (**b**) Cell viability in the HTB-132 cell line assay. Bar plots illustrate the percentage of cell viability after exposure to varying concentrations (2, 3, 25, and 100 μg/mL) of the venoms. (**c**) Cell viability of the PC3 cell line assay. Bar plots illustrate the percentage of cell viability after exposure to varying concentrations (2, 3, 6, 13, 25, 50, and 100 μg/mL) of the venoms. Bars denote ± standard error. Statistical significance compared with the control is indicated by (*) symbols. *M. helleri* (orange), *M. medemi* (green), and *M. sangilensis* (red) venoms.

**Table 1 toxins-15-00622-t001:** Relative abundances of the fractions obtained by the separation of whole venoms of *M. helleri*, *M. medemi,* and *M. sangilensis* by RP-HPLC on a C18 column. Lines depicting the three sections used to analyze chromatograms: 0 to 38 min, 38 to 50 min, and above 50 min. The most abundant components in each section are shown in bold and underlined.

Fraction	Retention Time	Relative Abundance		
		*M. helleri*	*M. medemi*	*M. sangilensis*
1	14.3	0.9	2.7	3.8
2	14.6	0.0	1.0	1.5
3	28.0	1.5	0.0	0.0
4	29.7	1.4	0.0	0.0
5	30.7	2.5	0.0	0.0
6	31.6	0.0	2.8	** 8.1 **
7	32.3	1.4	** 3.7 **	0.0
8	33.4	0.1	2.7	0.0
9	35.1	1.2	2.3	7.3
10	36.2	4.3	0.0	0.0
11	37.3	** 4.8 **	1.0	0.0
12	38.3	0.0	1.8	1.0
13	39.3	0.0	0.2	** 5.7 **
14	40.8	0.4	** 15.9 **	0.0
15	42.0	0.2	0.0	0.0
16	43.7	** 7.9 **	0.0	0.0
17	44.3	0.0	5.8	3.4
18	45.4	1.9	0.0	0.2
19	46.8	4.4	0.0	0.7
20	47.5	0.0	7.2	0.1
21	49.0	0.8	0.0	0.0
22	49.8	0.0	0.9	0.0
23	50.2	0.0	0.0	3.3
24	51.5	0.0	2.2	2.3
25	52.1	0.4	1.4	0.0
26	53.7	7.0	** 12.6 **	14.9
27	56.1	2.0	0.0	0.0
28	57.2	0.0	12.2	0.0
29	58.4	0.0	1.2	** 23.3 **
30	59.8	** 35.9 **	0.9	1.9
31	61.2	0.0	11.6	0.0
32	62.6	0.0	0.0	3.6
33	65.2	17.1	0.0	9.3
34	67.1	0.0	9.9	8.6
35	68.8	3.7	0.0	0.0
36	71.8	0.0	0.0	1.1

**Table 2 toxins-15-00622-t002:** Comparison of LD_50_ values from *Micrurus* venoms in this study compared with that of other Colombian *Micrurus* species. Values are given in μg/mice.

Species	LD_50_ (μg/Mice)	CI95%	Source
*M. dumerilli*	23.72	18.10–31.10	[41]
*M. helleri* (*sensu lato*)	22.87	17.79–29.40	
*M. isozonus*	6.29	5.14–7.69	
*M. mipartitus*	33.62	26.10–43.20	
*M. spixii*	13.89	Not determined	
*M. surinamensis*	29.17	24.42–34.84	
*M. helleri*	22.87	17.79–29.4	
*M. medemi*	8.79	7.16–10.80	
*M. sangilensis*	15.85	11.50–21.70	This study

Note: All assays were conducted via intraperitoneal injection (see Section 5 for details). CI: 95% confidence intervals. “Not determined” means that only death ratios corresponding to 0% and 100% were recorded for each trial.

## Data Availability

Not applicable.

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
