# Peer review of "Unveiling the Venom Composition of the Colombian Coral Snakes Micrurus helleri, M. medemi, and M. sangilensis"

_toxins, 2023, doi:10.3390/toxins15110622_

Round 1

Reviewer 1 Report

The deeper analysis of venom complexity and its functionality provides valuable information for a better understanding of venom-induced pathology and development of snakebite therapies. Micrurus sp. are medically important snakes in South America. In this context, Colombia harbours a high diversity of coral snakes, but some of their venoms remains unexplored. Motivated by this, the authors presented an integrative study that combines biochemical and functional characterization of three poorly studied venoms from Colombia: M.  medemi, M. sangilensis and M. helleri. The proteomic profiles confirmed the biochemical pattern of Micrurus venoms, classically represented by high abundance of low-molecular toxins, such as 3FTx and PLA2s. All three venoms showed a typical PLA2-rich phenotype. Intriguingly, an unusual and high presence of metalloprotease was detected. This detailed biochemical analysis was expanded with enzyme and cytotoxicity assays. The authors highlighted these underexplored venoms as rich source of cytotoxic agents useful for design of future biotherapeutics. In summary, the present manuscript is novel, pertinent and provide valuable information in Toxinology field. However, authors should address some key points before its publication.

1. Line 5-6 … composition and functional depiction of poorly know Colombian coral snakes This sentence is confusing. I believe the term coral snake venoms is more appropriate in this context.

2. Line 14-15. The trend in chromatographic profiles were great relative abundances for large and hydrophobic. I do not understand well this sentence. Peaks corresponding to metalloproteases were significant. Please improve this sentence, because this contradicts the proteomic profile, where PLA2 and 3FTxs are the most abundant toxins.

3. Line 17. Define best enzyme activity. This is not clear for me.

4. Line 26. The results showed a significant relative abundance of metalloprotease, but I do not agree that this turn the venoms rich in proteases. They are still rich in PLA2 and 3FTxs. Thus, please improve this sentence.

5. Figure1. Include units of molecular weight.

6. Please insert the gradient program in the chromatogram.

7. What was the criteria of selection of the fractions for SDS/PAGE analysis? Why did the authors not submit all fractions to electrophoresis?

8. Table 1. Please substitute , by .

9. Line 155. Please include more references. The authors mentioned studies by only one reference is supporting the sentence.

10. All enzymatic results can be presented in only one figure. Please combined all enzymatic data.

11. Figure 6 b. How many times was this experiment perform? Please include standard deviation.

12. Figure 7. Include units of molecular weight.

13. Figure 7. Change , by .

14. Line 235. This is table 2. There are two Tables 1 in the current version of the manuscript.

15. Page 11-12. Combine the cell viability results in only one figure.

16. Line 284. Please avoid one sentence-paragraph. Expand the central idea or combine with the next paragraph.  

17. Line 299-300. Error.  Reference source not found. Please add the correct reference.

18.  Line 363. From the Amazonian region of Colombia and Ecuador. Please add Ecuador, because the reference support this.

19. line 362-363. Please give more details to make this comparison clear. For example, add the percentage of the major toxins in the previous studies in comparison with your results.

20. Figures were mentioned and described in the results. Thus, I think it is not necessary to include Figures in the discussion section again.

21. Line 622. Cacl2: 2 should be a subscript

22. The introduction is long and a little unfocused. I recommend authors to summarize and focus on the state of the art important to their findings.

23. Given the interesting finding of a significant abundance of metalloprotease, the authors should discuss the role of metalloprotease in the context of Micrurus envenenomations. Comparisons with clinical cases and symptoms described should enrich the discussion.

Author Response

Please find the answers in the attached file.

Reviewer 2 Report

The details of the clotting experiments are not given in full. In particular, was the plasma recalcified? It is essential to add calcium back in so that spontaneous clotting is restored. Only then can coagulopathic activities be elucidated. It is impossible to judge the results of this study in regards to coagulation effects as absolutely no data is presented.  If the testing was done on plasma for which calcium was added back in, then the results must be presented as relative to the spontaneous clotting control. If the testing was not done on recalcified plasma, then either the testing must be redone or this section removed. In addition, Bothrops is a poor choice as the positive control as this genus has a procoagulant venom action mediated by snake venom metalloproseases, as opposed to the well-documented action by coral snakes of producing anticoagulation through phospholipid depletion, as per Dashevsky et al., 2020 Anticoagulant Micrurus venoms: Targets and neutralization. Toxicology Letters 337 (2021) 91–97.  As this study is measuring anticoagulation, an anticoagulant venom must be used as the positive control.

As a minor point, the authors point to reference 53 for evidence of thrombin-like activity. However, that article presents evidence free statements to this regard. Do the authors of the present work have a primary reference that clearly shows a thrombin-like effect? If not, then this statement must be removed. 

Author Response

(The authors gave the same response as above.)

Round 2

Reviewer 1 Report

The authors have clearly replied all of my concerns. In my opinion, the manuscript has been improved. The SDS-PAGE of all fractions would be very informative. However, the availability of coral snake venoms is a common limitation in many investigations. On the other hand, the funcional and proteomic characterization of these samples justify the publication of this study. 

Author Response

Dear Reviewer:

We sincerely appreciate your invaluable suggestions and meticulous review of the manuscript "Unveiling the Venom Composition of the Colombian Coral Snakes Micrurus helleri, M. medemi, and M. sangilensis."

We have diligently refined the manuscript to enhance the clarity and coherence of various sections, including the introduction, methods, results, and conclusions. The language has been meticulously polished to ensure precision and coherence throughout the document.

Enclosed is the revised paper, incorporating the modifications suggested during the review process. Notable modifications are highlighted in gray for convenience. All co-authors have meticulously reviewed and approved this manuscript version.

Once again, we extend our gratitude for your invaluable feedback, which has undoubtedly contributed to the enhancement of the manuscript, as your insights are pivotal to refining our work.

Thank you for your time and consideration.

Warm regards.

Reviewer 2 Report

The authors have not addressed the point as to whether calcium was added back into the plasma. If not, then the plasma remains in a physiologically deficient state as the entire point of adding citrate as a preservative, is strip out the calcium. Which then stops clotting from occuring due to the calcium-dependent nature of enzymes such as Factor Xa.  Thus, they need to provide details regarding their experimental design. It is of course impossible to note an anticoagulant effect (which has been shown for various Micrurus venoms) if the plasma does not spontaneously clot within normal parameters, which it will not do without the inclusion of calcium. If the authors did not use calcium, then their results are data-deficient and should be redone or dropped.

Author Response

Dear Reviewer:

We sincerely appreciate your invaluable suggestions and meticulous review of the manuscript "Unveiling the Venom Composition of the Colombian Coral Snakes Micrurus helleri, M. medemi, and M. sangilensis."

Considering your observations concerning the coagulant test conducted, we have decided to expunge any references to this assay from the methodology, results, and discussion sections. Regrettably, the assay cannot be replicated under suggested conditions due to the formidable challenge of acquiring larger quantities of venom.

Furthermore, we have diligently refined the manuscript to enhance the clarity and coherence of various sections, including the introduction, methods, results, and conclusions. The language has been meticulously polished to ensure precision and coherence throughout the document.

Enclosed is the revised paper, incorporating the modifications suggested during the review process. Notable modifications are highlighted in gray for convenience. All co-authors have meticulously reviewed and approved this manuscript version.

Once again, we extend our gratitude for your invaluable feedback, which has undoubtedly contributed to the enhancement of the manuscript, as your insights are pivotal to refining our work.

Thank you for your time and consideration.

Warm regards.

Round 3

Reviewer 2 Report

It is unfortunate that such deficient protocols are so widely used. This is a field-specific (venom research) issue largely due to historical contingency. Eg the Theakston  1983 protocol neglected to include it as it was designed without consultation with hematologists, who would have advised about the need to add calcium back in. This protocol is the one so often followed, leading to unfortunate cases here where the data quality was fine, it was just the protocol that was the problem. The authors have my sympathy.